

# Adaptive divergence, neutral panmixia, and algal symbiont population structure in the temperate coral *Astrangia poculata* along the Mid-Atlantic United States

Hannah E. Aichelman[1,2] and Daniel J. Barshis[2]

[1] Department of Biology, Boston University, Boston, MA, USA
[2] Department of Biological Sciences, Old Dominion University, Norfolk, VA, USA

Corresponding author
Hannah E. Aichelman,
haich@bu.edu

## ABSTRACT

*Astrangia poculata* is a temperate scleractinian coral that exists in facultative symbiosis with the dinoflagellate alga *Breviolum psygmophilum* across a range spanning the Gulf of Mexico to Cape Cod, Massachusetts. Our previous work on metabolic thermal performance of Virginia (VA) and Rhode Island (RI) populations of *A. poculata* revealed physiological signatures of cold (RI) and warm (VA) adaptation of these populations to their respective local thermal environments. Here, we used whole-transcriptome sequencing (mRNA-Seq) to evaluate genetic differences and identify potential loci involved in the adaptive signature of VA and RI populations. Sequencing data from 40 *A. poculata* individuals, including 10 colonies from each population and symbiotic state (VA-white, VA-brown, RI-white, and RI-brown), yielded a total of 1,808 host-associated and 59 algal symbiont-associated single nucleotide polymorphisms (SNPs) post filtration. Fst outlier analysis identified 66 putative high outlier SNPs in the coral host and 4 in the algal symbiont. Differentiation of VA and RI populations in the coral host was driven by putatively adaptive loci, not neutral divergence (Fst = 0.16, $p$ = 0.001 and Fst = 0.002, $p$ = 0.269 for outlier and neutral SNPs respectively). In contrast, we found evidence of neutral population differentiation in *B. psygmophilum* (Fst = 0.093, $p$ = 0.001). Several putatively adaptive host loci occur on genes previously associated with the coral stress response. In the symbiont, three of four putatively adaptive loci are associated with photosystem proteins. The opposing pattern of neutral differentiation in *B. psygmophilum*, but not the *A. poculata* host, reflects the contrasting dynamics of coral host and algal symbiont population connectivity, dispersal, and gene by environment interactions.

## INTRODUCTION

Population connectivity in marine systems is shaped by a complex mixture of factors, including oceanographic currents (*Selkoe, Henzler & Gaines, 2008*), planktonic larval durations and behavior (*Selkoe & Toonen, 2011*), life history dynamics (*Bradbury et al., 2008*), and environmentally driven selection (*Limborg et al., 2012*), among others

(reviewed in *Selkoe & Toonen, 2011*). The result of these interacting forces can manifest in numerous ways, including wide-ranging panmixia across 1,000s of kilometers (*Dao et al., 2015*), extraordinarily fine-scaled neutral population structure across 10s of meters (*Aurelle et al., 2011*; *Costantini, Fauvelot & Abbiati, 2007*; *Ledoux et al., 2010*), or local adaptation over both small and large spatial scales (*Bay & Palumbi, 2014*; *Bradbury et al., 2010*). Patterns of population connectivity in marine systems are made only more complex when considering organisms as multi-organism symbiotic communities, or holobionts, and each member of the holobiont can be subject to opposing forces driving population structure.

In scleractinian corals, the holobiont consists of the coral host, algal symbiont (family Symbiodiniaceae; *LaJeunesse et al., 2018*), and associated microbiota (*Knowlton & Rohwer, 2003*; *Rohwer et al., 2002*). While patterns are often complicated by species and location, population structure of the coral host is generally affected by reproductive strategy, with broad scale dispersal usually more common in broadcast spawning species compared to brooding species (*Ayre & Hughes, 2000*; *Coelho & Lasker, 2016*). Patterns of differential connectivity of coral hosts and symbionts has also been observed, with coral hosts generally exhibiting connectivity across larger scales compared to their hosted algal symbionts (*Baums, Devlin-Durante & LaJeunesse, 2014*; *Pettay & LaJeunesse, 2013*; *Pinzon & LaJeunesse, 2011*). This difference in dispersal among symbiotic partners could be influenced by distinct life history strategies and resulting dispersal abilities, transmission strategy of symbionts (i.e., horizontal vs. vertical), and/or differential selection pressures (*Baums, Devlin-Durante & LaJeunesse, 2014*).

Temperature is one environmental factor that can impose strong selection pressure on natural populations and therefore drive population differentiation (*Angilletta, 2009*), and is a common focus in the coral literature. Temperature gradients as drivers of selection and local adaptation have been demonstrated in corals, including *Porites astreoides* across nearshore and forereef environments on the Florida Keys Reef Tract (*Kenkel et al., 2013*; *Kenkel & Matz, 2016*) and *P. lobata* inhabiting tidal pools in American Samoa with differing patterns of thermal variability (*Barshis et al., 2018*; *Barshis et al., 2010*). While we are only beginning to understand these dynamics in tropical corals, comparatively little is known about the forces that shape population structure in temperate scleractinian corals, which exist across vastly wider environmental gradients. However, work in *Oculina* spp. has demonstrated genetic differentiation of the host did not correlate with Symbiodiniaceae community composition; instead, symbiont diversity and geographical structuring was shaped most strongly by sea surface temperature, particularly within the Mediterranean (*Leydet & Hellberg, 2016*). Additionally, work on temperate octocorals in the Mediterranean has demonstrated local adaptation to distinct temperature regimes across depths (*Haguenauer et al., 2013*; *Ledoux et al., 2015*) while highlighting genetic drift in limiting such phenotypic differentiation (*Crisci et al., 2017*; *Ledoux et al., 2015*).

Here, we consider patterns of neutral and adaptive differentiation in the northern star coral, *Astrangia poculata* (= *A. danae*; *Peters et al., 1988*), and its algal symbiont *Breviolum psygmophilum* (*LaJeunesse, Parkinson & Reimer, 2012*). *Astrangia poculata* is

facultatively symbiotic, meaning it can exist both with (symbiotic or brown) or without (aposymbiotic or white) *B. psygmophilum* across its range (*Boschma, 1925*). Therefore, the forces shaping host population structure may be decoupled from those shaping the algal symbiont. Our previous work (*Aichelman, Zimmerman & Barshis, 2019*) demonstrated physiological differentiation between a Virginia (VA) and Rhode Island (RI) population of *A. poculata*. Namely, the thermal optimum ($T_{opt}$) of respiration was elevated in VA corals compared to RI (3.8 °C and 6.9 °C greater in brown and white corals, respectively), corresponding with warmer in situ temperatures in the more southern population and suggestive of a signature of adaptation of VA and RI corals to their native thermal habitat.

Studies from Narragansett Bay, RI have demonstrated that *A. poculata* is a gonochoristic broadcast spawner that releases its gametes into the water column between early August and September (*Szmant-Froelich, Yevich & Pilson, 1980*). *A. poculata* horizontally transmits its symbionts, and recruits therefore acquire symbionts locally once they have settled in their new environment (*Szmant-Froelich, Yevich & Pilson, 1980*). Original studies on *A. poculata* life history observed the development of the planula larval stage 12–15 h after fertilization, but researchers were unable to induce settlement (*Szmant-Froelich, Yevich & Pilson, 1980*). Achieving larval settlement in the lab remains challenging in this species today; however, larvae have been observed to remain swimming for at least 5 weeks in the lab (D. Wuitchik, 2019, personal communication), suggesting an extended pelagic larval duration (PLD).

In the current study, we characterized population structure of the *A. poculata* host and its symbiont (*B. psygmophilum*) both by population (VA vs. RI) and by symbiotic state (brown vs. white) using SNPs derived from high-throughput mRNA-Seq data. Leveraging our previous work on physiological adaptation in these populations, we explore the relative strengths of neutral and adaptive divergence in this broadcast spawning, facultatively symbiotic, and extraordinarily thermally resilient species across the northern half of its range.

## MATERIALS AND METHODS

### Sample collection

Corals were collected from the same sites and using the same methods as previously described (*Aichelman, Zimmerman & Barshis, 2019*). Here, in August and September, 2017, brown and white *Astrangia poculata* colonies were collected each from Virginia (VA) and Rhode Island (RI), USA (populations will be referred to as VA-brown ($n = 10$), VA-white ($n = 10$), RI-brown ($n = 10$), and RI-white ($n = 10$)). All brown and white VA colonies were collected from the wreck of the *J.B. Eskridge* (36°53′57.1″N, 75°43′20.6″ W) on September 15, 2017 at 20 m depth (Fig. 1). All VA colonies were transported to the Old Dominion University (ODU) Aquatics Facility within 6 h of collection. All brown and white RI colonies were collected from Fort Wetherill State Park (41°28′ 38.7″N, 71°21′36.3″W) on August 3, 2017 from a depth of 11 m (Fig. 1). RI colonies were maintained overnight submerged in Narragansett Bay and then transported by car in an aerated aquarium to ODU the next day. All VA and RI colonies were collected using a

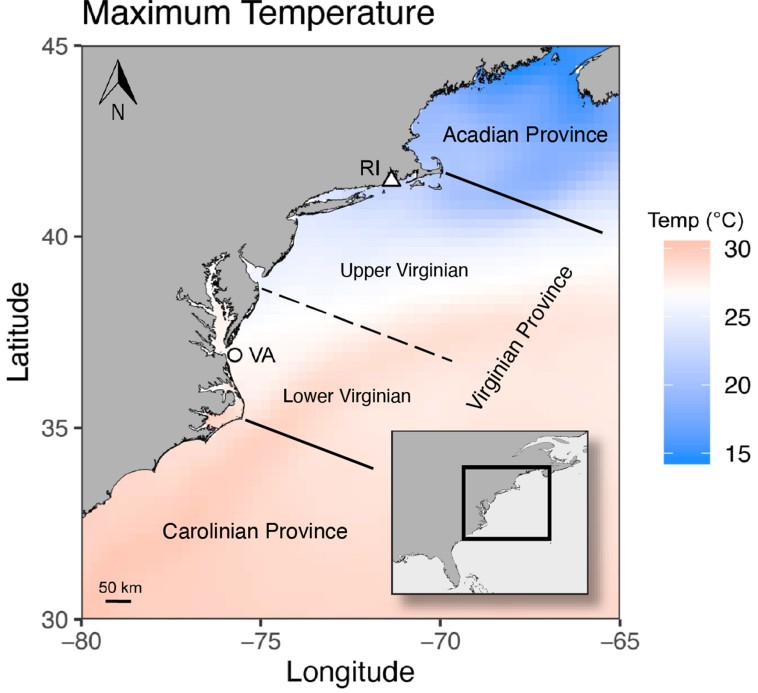

**Figure 1 Maximum sea surface temperatures averaged across 1982 to 2018 for part of the range of *Astrangia poculata*, including the populations sampled in this study.** Virginia = VA and Rhode Island = RI. The over-water distance between the two collection sites is approximately 630 Km. Biogeographic province designations are from *Briggs (1974)* and *Engle & Summers (1999)*.

hammer and chisel and were separated by at least 0.5 m to ensure the collection of distinct individuals. VA corals were collected under Virginia Marine Resources Commission permit #17-017, while no permit was required for collection of RI corals.

Following both collections, all *A. poculata* colonies were placed in a holding aquarium at ODU and maintained at a temperature of 18 °C (±0.5 °C) and salinity of approximately 35 ppt for a recovery period before fragmentation (5 days for VA corals, 48 days for RI corals). Corals were fragmented over 2 days (September 20–21, 2017) using a high-speed cut off tool (Chicago Pneumatic, Rock Hill, SC, USA) fitted with a diamond tip circular blade (sensu *Aichelman, Zimmerman & Barshis, 2019*), then affixed to labeled underwater paper using InstaCure ethyl cyanoacrylate gel (IC-Gel; Bob Smith Industries Inc., Atascadero, CA, USA). All individuals (i.e., putative genets) were fragmented into three pieces, so that a ramet from each genet was represented in each of the three temperature treatments (cold = 14 °C, control = 18 °C, and heat = 22 °C). Following fragmentation, all corals were allowed to recover at the holding conditions (18 °C and 35 ppt) for 20 days before the experiment began. While this experiment was designed to consider differential gene expression in *A. poculata* across temperatures and populations; here, we present sequencing data from this same experiment and resulting single nucleotide polymorphisms (SNPs) to consider population structure.

## Temperature environment of coral populations

To compare long-term temperature trends across the two collection sites, sea surface temperature (SST) data was downloaded from the NOAA 1/4° daily Optimum Interpolation Sea Surface Temperature dataset (OISST, dataset ID=ncdcOisst2Agg_LonPM180; *Banzon et al., 2020*) using the griddap function implemented in R (v3.5.2). Data was downloaded for the years 1982–2018 spanning the area 30–45° latitude and −80° to −60° longitude. Annual maximum temperature (Fig. 1), annual mean (Fig. S1A), and annual minimum temperature (Fig. S1B) of each 1/4° pixel were calculated from this dataset. The OISST temperature data were downloaded, compiled, and plotted with a high-resolution shoreline map layer (*Wessel & Smith, 1996*) using the method detailed in the GitHub repository associated with this publication.

## Aquaria conditions

Similar to *Aichelman, Zimmerman & Barshis (2019)*, following collection, all experimental corals were maintained in a 325-gallon holding aquarium with artificial seawater mixed using Crystal Sea® Bioassay salt (Marine Enterprises International, Baltimore, MD, USA) and deionized (DI) water. Temperature was maintained using a temperature controller (AquaLogic, San Diego, CA, USA) in combination with an in-line water chiller (Delta Star®, AquaLogic, San Diego, CA, USA) and 1,500 W immersion heater (Process Technology, Willoughby, Ohio, USA). The holding aquarium was equipped with a filter sock for mechanical filtration, protein skimmer for removal of organic material, and powerheads (Tunze® Turbelle, Penzburg, Germany) to maintain flow. All experimental corals were fed three times a week with freshly hatched *Artemia* sp. nauplii and maintained under approximately 200 µmol photons m$^{-2}$s$^{-1}$ of light supplied by 165 W LED aquarium lights (GalaxyHydro, Roleadro, Shenzhen, China).

   The experiment was run in three separate aquaria, one for each temperature treatment (cold = 14 °C, control = 18 °C, and heat = 24 °C). As in the holding aquarium, artificial seawater was mixed to 35 ppt and each aquarium was equipped with a filter sock and protein skimmer. In the heat and cold aquaria, temperature was manipulated using a custom-programmed Arduino® (code assembled by D. Barshis) connected to the same style heater and chiller as the holding aquarium (sensu *Aichelman, Zimmerman & Barshis, 2019*). Temperature in the control experimental aquarium was maintained using a temperature controller as in the holding aquarium (AquaLogic, San Diego, CA, USA).

## Experimental design

At 08:00 on October 12, 2017, all corals were moved from the holding tank to one of the three experimental aquaria, such that each genotype was represented in each temperature treatment. Each experimental aquarium was maintained at conditions identical to the holding tank (salinity = 35 ppt, temperature = 18 °C). All corals were given 30 min of acclimation in the dark, after which the lights were turned on and a 1 h hold at 18 °C began. At 09:30, temperature was increased (heat aquarium) and decreased (cold aquarium) at a rate of 4 °C h$^{-1}$, and the target temperatures of 22 °C (heat) and 14 °C

(cold) were reached at 10:30. The corals were held at their respective target temperatures for another hour, and the experiment therefore ended at 11:30 (Fig. S2).

Immediately upon completion of the experiment, tissue samples of approximately 1 cm$^2$ were taken from each coral fragment and preserved in an RNAlater-like solution (*De Wit et al., 2012*). RNA sampling took place between 11:30 and 13:10, and all fragments were maintained at their respective experimental condition until sampling occurred. The tissue samples were stored in RNALater at −80 °C until RNA extraction.

Temperature was monitored throughout the experiment using Hobo Pendant® Temperature Data Loggers (Onset Computer Corporation, Bourne, MA, USA; Fig. S2), which recorded water temperature at 1-min intervals. Temperature loggers were calibrated with a NIST certified glass thermometer. Light levels in each experimental aquarium were maintained at 410 µmol photons m$^{-2}$s$^{-1}$ during the experiment, as measured in the center of each aquaria and supplied by a 165 W LED aquarium light (GalaxyHydro, Roleadro, Shenzhen, China). The light levels used here were based on an estimation of minimum saturating irradiance from our previously published photosynthesis vs. irradiance curve (*Aichelman, Zimmerman & Barshis, 2019*) as well as previous work by *Jacques, Marshall & Pilson (1983)*.

## Library preparation and sequencing

RNA was extracted from all coral fragments between November 1 and 13, 2017 using Trizol (Invitrogen, Carlsbad, CA, USA). For each extraction, coral tissue samples were thawed, crushed with a sterile razor blade, combined with Trizol and sterile 0.5 mm zirconia/silica beads (BioSpec Products, Bartlesville, OK, USA) and bead beat on high for 2 min to break open cells. Following a 5-min incubation at room temperature, samples were centrifuged for 10 min at 12,000×$g$ and 4 °C to pellet the skeleton and beads. The supernatant was combined with 200 µl of chloroform and mixed for 15 s. Following another 5-min incubation, samples were spun for 15 min at 12,000×$g$ and 4 °C, and the chloroform cleaning step was repeated. After the last centrifugation step, the supernatant was combined with an equal volume of 95% ethanol and all samples were immediately purified using a Direct-zol RNA MiniPrep Plus purification kit (Zymo Research, Irvine, CA, USA) according to manufacturer's instructions. The purified RNA was eluted in 50 µL of DNase/RNase-free water and immediately stored at −80 °C for future library preparation. The concentration of all RNA extractions was assessed using a Qubit 2.0 Fluorometer (Invitrogen by Life Technologies, Carlsbad, CA, USA) and diluted to 0.1–1 µg total RNA before mRNA-Seq library preparation.

Libraries ($N = 96$) were prepped using the Illumina TruSeq mRNA prep kit (San Diego, CA, USA) and half-sized reaction volumes. The 96 libraries consisted of 7 genotypes per population (RI-brown, RI-white, VA-brown, VA-white), with each genotype represented across all temperature treatments (7 genotypes × 4 populations × 3 temperature treatments = 84 libraries). These 84 libraries were intended for use in differential gene expression analysis. An additional 12 libraries, intended to increase the sample size for SNP analyses, were prepared from 3 additional genotypes per population

(3 genotypes × 4 populations = 12 libraries), all of which were in the 18 °C control treatment.

The quality and quantity of all mRNA-Seq libraries was assessed using both a fragment analyzer (DNF-910 dsDNA Reagent Kit, Agilent Technologies, Santa Clara, CA, USA) and KAPA Library Quantification Kit for Illumina platforms (Roche Sequencing Solutions, Pleasanton, CA, USA). Any libraries that did not successfully amplify after the first attempt were prepped again using the same method listed above, except 25 μL of un-diluted RNA was used in the Illumina TruSeq mRNA prep kit. The 96 libraries were sequenced on 6 lanes of an Illumina HiSeq4000 (16 libraries per lane) at the University of California Berkeley Vincent J. Coates Genomics Sequencing Laboratory, which yielded single-end 50 base pair (bp) reads. All RNA sequences have been deposited in NCBI BioProject under accession number PRJNA614998.

## Processing sequences and transcriptome assembly

Detailed descriptions for all data analyses can be found on the electronic notebook associated with this publication (github.com/hannahaichelman/Astrangia_PopGen). Raw sequences were processed using the adapter trimming/quality filtering functions of the Fastx-Toolkit to remove adapter sequence contamination and reads with a quality score less than 33. All sequences were then used as input for de novo transcriptome assembly using Trinity (version 2.0.6; *Grabherr et al., 2011*) and default parameters. Ribosomal RNA (rRNA) contamination of the reference was identified using nucleotide blast (blastn) against the Silva large subunit (LSU) and small subunit (SSU) databases (http://www.arb-silva.de/). "Good hits" to these rRNA databases were defined as matching at least 78% of the read over at least 100 bp, and once identified were removed from the reference assembly. A total of 1,273 matches to the LSU database and 688 matches to the SSU database were removed from the reference assembly.

Once rRNA contamination was removed from the reference assembly, it was filtered to include only sequences greater than 500 bp in length. Host and symbiont contigs in the reference assembly were differentiated and assigned as described previously (*Barshis et al., 2013*; *Davies et al., 2016*; *Ladner & Palumbi, 2012*). Briefly, the reference assembly was blasted (using blastn) against four databases: (1) all available cnidarian data ("dirty coral"; $n = 10$ datasets), (2) all available Symbiodiniaceae data ("dirty sym"; $n = 8$ datasets), (3) aposymbiotic cnidarian data only ("clean coral"; $n = 15$ datasets), and (4) cultured Symbiodiniaceae data only ("clean sym"; $n = 9$ datasets). These four databases contained the same sequencing data used by *Davies et al. (2016)*, plus an additional 16 datasets (summarized in Table S1). A contig was considered a host contig if it had a length overlap greater than 100 bp with a 60% identity cutoff to any cnidarian. If the same contig was also assigned to a cultured (clean) symbiont read with the same length and cutoff identity, it was removed from the host contig list. Similarly, a contig was considered a symbiont contig if it had a length overlap greater than 100 bp with a 60% identity cutoff to any symbiont, and removed if it also assigned to a clean coral reference. Contigs identified as both coral and symbiont were also removed from the reference.

**Table 1 Summary statistics for the *Astrangia poculata* and *Breviolum psygmophilum* combined reference assembly constructed using Trinity.**

| | |
|---|---|
| Total number of sequences | 15,079 |
| Average sequence length | 876 bp |
| N50 | 881 |
| Median sequence length | 578 bp |
| Percent annotated | 91.7% |

Once contigs were designated as host or symbiont, the resulting Trinity-assembled reference was annotated by BLAST sequence homology searches against GenBank's nr protein database (*Agarwala et al., 2018*) and UniProt's Swiss-Prot and TrEMBL databases (*The UniProt Consortium, 2018*) to create a consensus annotation using an *e*-value cutoff of $1e^{-4}$ using a custom annotation script (*De Wit et al., 2012*). Annotated sequences were then assigned to Gene Ontology (GO) categories (*The UniProt Consortium, 2018*). Reference transcriptome size and contiguity was assessed using a custom python script (Table 1).

## SNP calling and clone identification

To maximize the total reads per genotype for the population genetic analyses presented here, fastq files were combined by genotype. This reduced 84 libraries (7 genotypes × 4 populations × 3 temperature treatments) to 28 fastq files. These concatenated files, along with the 12 libraries originally intended for population genetic analyses, yielded a total of 40 libraries that were used for all downstream analyses. It should be noted that the dataset presented here had relatively low read counts, resulting from an error in the library preparation that led to sequencing of rRNA in addition to mRNA. The primary result of this error was a larger than usual percentage of sequence yield going to rRNA, which resulted in a lower read count on an individual basis. To combat this, we have used conservative cut-off values at every step of the data analysis pipeline to address these issues and account for missing data.

The quality filtered reads were mapped to the de novo holobiont transcriptome using the "very-sensitive" method of Bowtie2.2.4 (*Langmead & Salzberg, 2012*). Single nucleotide polymorphisms (SNPs) were detected from the mapped reads using freebayes (*Garrison & Marth, 2012*). The resulting unfiltered variant call file (vcf) was separated into "good coral SNPs" and "good symbiont SNPs" using the lists of coral and symbiont contigs in the reference transcriptome. These coral and symbiont vcf files were then separately filtered using vcftools (v0.1.12b) and utilizing scripts from the dDocent/2.24 pipeline (*Puritz, Hollenbeck & Gold, 2014*; *Puritz et al., 2014*) to create a rigorously filtered set of variant sites. The filtering parameters were the same for the host and symbiont SNPs, and each file was filtered in four steps. First, vcftools was used to filter files to exclude individuals with more than 50% missing data, exclude sites with minor allele count (mac) greater than or equal to 3, only include sites with a quality score above 30, exclude genotypes with fewer than 5 reads, include only bi-allelic sites, and remove indels. Second,

the filter_missing_ind.sh script from dDocent was used to remove individuals with more than 85% missing data. Third, another round of vcftools filtering was conducted to exclude sites if they had more than 75% missing data, include sites with a minor allele frequency (maf) greater than or equal to 0.05, and include sites with mean depth values (across all included individuals) greater than or equal to 10. After these filters were executed, the fourth filtering step used the filter_hwe_by_pop.pl script from dDocent to remove sites out of Hardy-Weinberg equilibrium within each population (minimum Hardy–Weinberg $p$-value cutoff for SNPs = 0.01). This fourth filter resulted in no SNPs being removed from the symbiont file, but 11 SNPs being removed from the coral file. The filtered vcf files were converted to genepop format using a custom python script (written by D. Barshis). The coral host analysis was conducted considering the data as four populations (VA-B, VA-W, RI-B, and RI-W). The symbiont data only included brown hosts and was therefore analyzed as two populations (VA-B and RI-B). Additionally, to look for consistent loci driving patterns of differentiation between brown and white phenotypes within VA and RI, the coral host analysis was also conducted as two populations for each origin separately (VA-B and VA-W separately from RI-B and RI-W).

Potential clones in the coral host data were assessed using the R package poppr (v2.8.5; *Kamvar, Brooks & Grünwald, 2015*; *Kamvar, Tabima & Grünwald, 2014*). The filtered vcf file was imported into R and converted into the genind format. Then, the diss.dist function was used to calculate a distance matrix based on relative dissimilarity (i.e., Hamming's distance; *Hamming, 1950*; *Wang, Kao & Hsiao, 2015*), or the number of allelic differences between two individuals. The distance matrix was clustered with hclust and a dendrogram was plotted to identify clones. Two RI-B individuals were determined to be clones using this method, and one was randomly removed for all downstream analyses (Fig. S3). To ensure that the clone identification was robust to other measures of genetic distance, this method was repeated with Prevosti distance (*Prevosti, Ocana & Alonso, 1975*) and Manhattan distance (using the vegdist function in the package vegan v2.4-2; *Oksanen et al., 2011*), and the same pair of clones was detected each time.

## Outlier detection

Neutral and outlier SNPs were detected using BayPass3.04 (*Gautier, 2015*) under the core model. Outlier SNPs under the core model were determined based on the XtX statistic (introduced by *Günther & Coop (2013)*), which is a SNP-specific Fst corrected for the scaled covariance of population allele frequencies (*Gautier, 2015*). Outlier SNPs were defined as having XtX > 0.5% FDR from a simulated dataset. The results of the BayPass analysis were used to create a neutral and a high outlier list of SNPs, and these two sets were analyzed separately for all downstream analyses. Total numbers of neutral and outlier SNPs are summarized in Table 2.

## Population structure and genetic diversity analyses

Following BayPass outlier detection, the coral and symbiont vcf files were each filtered to create separate files for neutral and high outlier SNPs. These separate vcf files were then randomly filtered to include only one SNP per contig using the

**Table 2 Summary of number of SNPs.**

| Analysis Step | Host | Symbiont |
|---|---|---|
| *(A) Filtered vcf File* | | |
| Total number of SNPs | 1,808 | 59 |
| High Outlier SNPs | 84 | 4 |
| Neutral SNPs | 1,637 | 52 |
| *(B) Filtered 1 SNP per Contig File* | | |
| High Outlier SNPs | 66 | 4 |
| Neutral SNPs | 279 | 20 |

Note:
Summary of the number of SNPs after original filtering using vcftools (A) and after filtering to include only one SNP per contig as in the dDocent pipeline (B). Filtering for 1 SNP per contig was done after original filtering using vcftools and separating neutral and high outlier loci. Number of SNPs are included for the host (VA-B, VA-W, RI-B, RI-W) and symbiont (VA-B and RI-B) analyses discussed here.

Filter_one_random_snp_per_contig.sh script from the dDocent pipeline (*Puritz, Hollenbeck & Gold, 2014*; *Puritz et al., 2014*). This was done to avoid potential bias of analyses due to non-independence of SNPs on the same contig. All analyses besides BayPass outlier detection used files that were first separated by neutral and outlier SNPs and then filtered to include only one SNP per contig.

Pairwise differentiation (Fst) calculations were conducted in GenoDive v3.04 (*Meirmans & Van Tienderen, 2004*) using the AMOVA Fs method and tested with 999 permutations. AMOVA analyses were conducted in GenoDive using the infinite allele model (Fs-analog), a structure of allele nested within individual nested within population, and tested with 999 permutations. Measures of genetic diversity per population, including observed heterozygosity (Ho) and heterozygosity within subpopulations (i.e., expected heterozygosity; Hs) were calculated in GenoDive using default settings. To visualize population differentiation, Principal Component Analysis (PCA) was conducted using the R (v3.5.2) package *Adegenet* (*R Core Team, 2017*).

STRUCTURE (v2.6), with a 100,000 burn-in period and 500,000 MCMC runs after burn-in, was used to determine the number of distinct genetic groups in the coral host (*Falush, Stephens & Pritchard, 2007*). The optimal number of genetic clusters (*K*) was assessed using the DeltaK method (*Evanno, Regnaut & Goudet, 2005*), which was computed via the online program StructureSelector (*Li & Liu, 2018*) and plotted (Fig. S4) using the online program Clumpak (*Kopelman et al., 2015*).

## Gene Ontology enrichment of outlier SNPs

A gene ontology enrichment analysis was conducted to determine if genes containing outlier SNPs were enriched for any particular GO categories. Each gene in the reference transcriptome received a binary indicator of whether it contained an outlier SNP or not, and a GO enrichment analysis based on Fisher's exact test was used to determine if these outlier SNPs were enriched for any GO categories. Results were plotted as a dendrogram (Fig. S5), which indicates gene sharing between GO categories and lists the number of genes in the module over the total number of genes assigned to each category in the entire SNP dataset (*Davies et al., 2016*; *Dixon et al., 2015*).

**Table 3 Summary of sequencing and mapping results.**

| Filter type | Number of reads |
| --- | --- |
| After merging files by genotype | 9,340,966–167,484,549 (51,082,950.3 ± 4,634,637.5) |
| Singly aligned mapped | 4,720,063–85,431,605 (25,860,167.1 ± 2,361,454.9) |
| % Singly aligned mapped | 45.96–54.1% (50.6 ± 0.26%) |

Note:
Each category contains the range followed by the average ± standard error in parentheses.

**Table 4 Pairwise differentiation (Fst) results from Genodive for the *Astrangia poculata* host.**

| | VA-B | VA-W | RI-B | RI-W |
| --- | --- | --- | --- | --- |
| VA-B | – | – | – | – |
| VA-W | −0.007 (0.842) | – | – | – |
| | **0.094 (0.001)** | | | |
| RI-B | 0.008 (0.159) | −0.003 (0.649) | – | – |
| | **0.213 (0.001)** | **0.189 (0.001)** | | |
| RI-W | **0.012 (0.026)** | 0.002 (0.377) | 0.003 (0.325) | – |
| | **0.196 (0.001)** | **0.181 (0.001)** | **0.079 (0.002)** | |

Note:
Virginia = VA and Rhode Island = RI, B = symbiotic host, W = aposymbiotic host. The first number reported is the Fst value and the number following in parentheses is the *p*-value (significance was tested using 999 permutations in Genodive). Significant pairwise differentiation results are bolded. In each box, the results on the top are for putatively neutral loci, and the results underneath are for high outlier loci.

# RESULTS

## Mapping and SNP numbers

The total counts of *A. poculata* mapped reads/genet ranged from 9,340,966 to 167,484,549, and mapping efficiencies ranged from 45.96% to 54.10% (Table 3). The unfiltered vcf file had a total of 1,214,003 variants, 432,676 of which were on host contigs and 16,417 of which were on symbiont contigs. After all filters were applied, there were a total of 1,808 coral SNPs, and 59 symbiont SNPs (two populations including brown individuals only; Table 2). After removing individuals with a high percentage of missing data from the coral vcf file ($n$ = 2 RI-B, $n$ = 2 RI-W, $n$ = 1 VA-B) as well as removing one putative clone (RI-B), a total of 34 individuals remained in the coral analysis ($n$ = 7 RI-B, $n$ = 8 RI-W, $n$ = 9 VA-B, $n$ = 10 VA-W). For the symbiont analysis, two individuals with a high percentage of missing data ($n$ = 2 VA-B) were removed, leaving 38 individuals in the symbiont analysis ($n$ = 10 RI-B, $n$ = 10 RI-W, $n$ = 8 VA-B, $n$ = 10 VA-W).

## Significant neutral population structure in symbiont, but not host

For the coral host analysis, BayPass analyses identified 1,637 neutral and 84 high outlier SNPs, and further filtering for one SNP per contig left 279 neutral loci and 66 high outlier loci that were used in population differentiation analyses (Table 2). Analysis of Molecular Variance (AMOVA) tests revealed no significant population structure at neutral loci (Fst = 0.002, AMOVA $p$ = 0.269); however, there was significant population structure at high outlier loci (Fst = 0.16, AMOVA $p$ = 0.001; Table 4). Pairwise differentiation

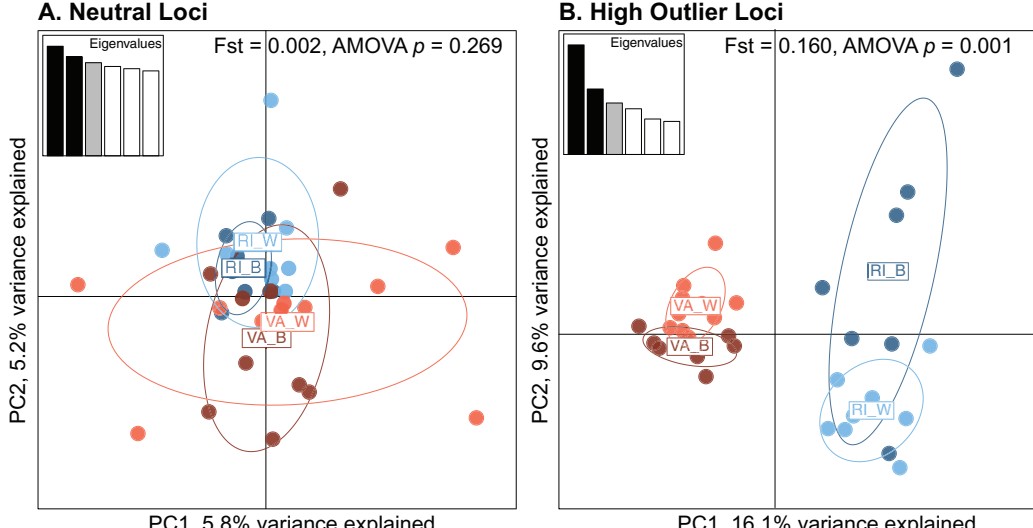

**Figure 2 Principal Component Analysis (PCA) of neutral (A, *n* = 279) and high outlier (B, *n* = 66)** ***Astrangia poculata* SNPs of the four populations.** Colors represent the four populations: dark orange = VA-B (brown/symbiotic Virginia), light orange = VA-W (white/aposymbiotic Virginia), dark blue = RI-B (brown/symbiotic Rhode Island), and light blue = RI-W (white/aposymbiotic Rhode Island). Fst values and associated *p*-values are from Analysis of Molecular Variance (AMOVA) tests implemented in Genodive. The *x*- and *y*-axes indicate the percent of the variance explained by the first and second principal component, respectively. Insets indicate the eigenvalues of the first six principal components.

between the four populations reflects a similar pattern, with no significant population structure at neutral loci for most pairwise comparisons (Fst ranges between −0.007 and 0.012, all *p* > 0.159; Table 4) except between VA-B and RI-W (Fst = 0.012, *p* = 0.026). At high outlier loci, all pairwise Fst values were significant (Fst ranges between 0.079 and 0.213, all *p* < 0.01; Table 4). Principal components analysis (PCA) further demonstrates that all four populations overlap at neutral loci (Fig. 2A) but separate primarily by origin when including only putatively adaptive loci (Fig. 2B).

For the symbiont analysis, BayPass analyses identified 52 neutral and 4 high outlier SNPs, and further filtering for one SNP per contig left 20 neutral loci and 4 high outlier loci that were used in population differentiation analyses (Table 2). In contrast to the coral host population differentiation, symbiont SNPs showed significant population structure at neutral loci (Fst = 0.093, AMOVA *p* = 0.001; Fig. 3). At the four high outlier symbiont SNPs identified, there was also significant differentiation (Fst = 0.965, AMOVA *p* = 0.001).

The coral host Fst results are reflected in the STRUCTURE analysis, with a pattern of admixture in all samples at neutral loci in the coral host, but two distinct clusters separating VA and RI individuals at high outlier loci (Fig. S4). The optimal K value for both neutral and outlier loci in the coral host was two.

## Genetic diversity

For the coral host analysis, at neutral loci there were no clear patterns for one population having higher observed (Ho) or expected (Hs) heterozygosity than another, with Ho

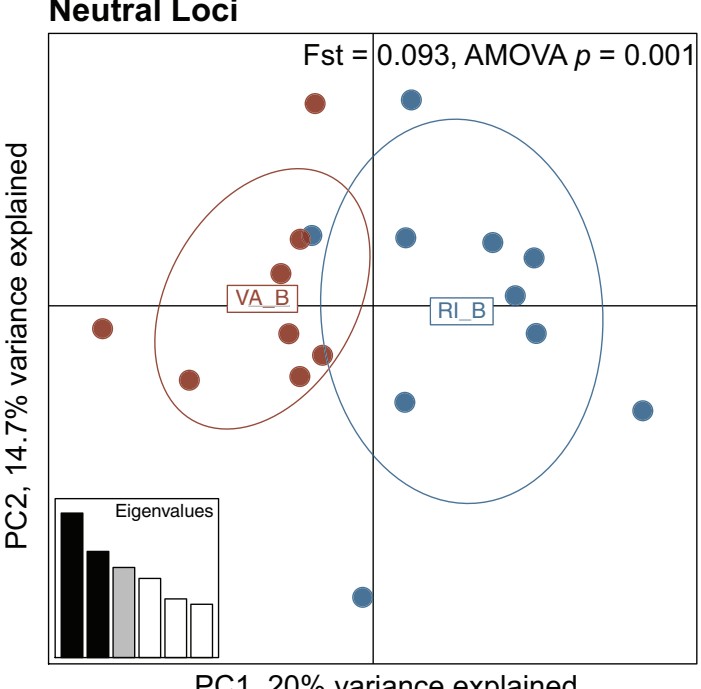

**Neutral Loci**

Fst = 0.093, AMOVA *p* = 0.001

*Figure 3* **Principal Component Analysis (PCA) of neutral *Breviolum psygomphilum* SNPs (*n* = 20).**
Colors represent the two populations: dark orange = VA-B (brown/symbiotic Virginia) and dark
blue = RI-B (brown/symbiotic Rhode Island). Fst, associated *p*-values, and labeling are as in Fig. 2.

ranging between 0.217 (RI-B) and 0.283 (RI-W) and Hs ranging between 0.224 (RI-B)
and 0.267 (VA-W) (Table S2). At high outlier loci for the coral host, Ho and Hs was
lower in VA populations (Ho: VA-B = 0.121, VA-W = 0.097; Hs: VA-B = 0.168,
VA-W = 0.171), than in RI (Ho: RI-B = 0.191, RI-W = 0.292; Hs: RI-B = 0.324,
RI-W = 0.283; Table S2). For the neutral symbiont loci, Ho and Hs were higher in the
VA-B population (0.575 and 0.385, respectively) than in the RI-B population (0.47 and
0.338, respectively). For the four high outlier symbiont SNPs, Ho and Hs were both 0.0 in
VA-B population, and in the RI-B population Ho = 0.050 and Hs = 0.047 (Table S2).

### Outlier SNP identities

Coral host contigs that contained more than one high outlier SNP (*n* = 13) were manually
inspected to look for agreement with previous studies on coral adaptation to distinct
temperature environments (Table S3). Some of these genes have been previously observed
in coral transcriptomic studies in response to thermal stress, including 40S and 60S
ribosomal proteins (*DeSalvo et al., 2010*; *Portune et al., 2010*), myosin heavy chain
(*Woo et al., 2010*), and ubiquitin-like protein FUBI (*Barshis et al., 2010*). Additionally,
one gene (apolipoprotein B-100) was previously implicated in playing a role in the
coral symbiosis (*Bertucci et al., 2015*). Two other genes were highlighted in previous
transcriptomic studies in corals, including the putative immune gene cyclic

AMP-dependent transcription factor ATF-5 (*Fuess, Weil & Mydlarz, 2016*) and two collagen chain proteins implicated in calcification and/or modified cell adhesion (*DeSalvo et al., 2010*).

There were four high outlier SNPs in symbiont reads, three of which were annotated. These SNPs are on genes annotated as photosystem II (PSII) CP43 reaction center protein, photosystem I (PSI) P700 chlorophyl *a* apoprotein A2, and PSII protein D1. A full summary of these contigs with multiple high outlier SNPs can be found in Table S3.

When analyzing the data as two populations separately for VA and RI to look for consistent high outlier SNPs that could be driving differentiation between brown and white morphs, 11 high outlier SNPs were identified as shared between VA and RI (Table S3), which independently had 49 and 47 high outlier SNPs, respectively. These shared high outlier SNPs include several previously mentioned genes (40S and 60S ribosomal proteins, cyclic AMP-dependent transcription factor ATF-5, and a collagen chain protein), as well as Sequestosome-1, which is implicated in autophagy and immune system process (UniProtKB entry 008623; *The UniProt Consortium, 2018*).

### High outlier SNPs enriched for ribosome-associated GO categories

Gene Ontology enrichment analyses showed that the coral host outlier SNPs detected in this study were enriched for 10 GO terms in the "molecular function" category and seven GO terms in the "cellular components" category. The significant cellular components terms consisted of several ribosome-associated terms, including large ribosomal subunit, small ribosomal subunit, ribosome, and ribosomal subunit. The significant molecular function terms consisted of several ribosome-associated terms, including structural constituent of ribosome and rRNA binding, as well as several terms related to coral stress response, including ferric iron binding and oxidoreductase (Fig. S5).

## DISCUSSION

### Opposing patterns of differentiation in *Astrangia poculata* hosts and symbionts

Here, we find evidence of contrasting patterns of population structure between members of the holobiont in the temperate scleractinian coral *Astrangia poculata*. Between Virginia and Rhode Island, the coral host exhibited neutral panmixia, but adaptive divergence. While there was no significant population differentiation at putatively neutral loci in the host, there was evidence of neutral divergence in the algal symbiont. *Astrangia poculata* is a gonochoric broadcast spawning species that horizontally transmits its symbionts, a life history strategy that likely lends to the contrasting patterns of genetic differentiation in the host and symbiont observed here. Marine broadcast spawning species have extended pelagic larval durations (PLDs) of up to 244 days (*Graham, Baird & Connolly, 2008*), facilitating long-range dispersal and genetic connectivity (*Ayre & Hughes, 2000*; *Davies et al., 2017*; *Nishikawa, Katoh & Sakai, 2003*). Combined with observations of *A. poculata* larvae swimming 5 weeks post-release in the lab (D. Wuitchik, 2019, personal communication), it is not unreasonable to expect that *A. poculata* larvae can

remain in the water column for extended periods of time, facilitating larval connectivity across VA and RI.

Previous work in corals has demonstrated population connectivity across great distances. For example, high genetic connectivity was found across 4,000 Km in Micronesia for two acroporid corals (*Davies et al., 2015*). In the Caribbean, models predict that the extended PLD (20–120 days) of *Orbicella franksi* enables export of larvae from the Flower Garden Banks to more distant Caribbean reefs, including Broward and Palm Beach, Florida approximately 1,500 Km away (*Davies et al., 2017*). Additionally, across 10 Caribbean populations (bounded by Barbados in the east, Belize in the west, and the Flower Garden Banks in the north), *O. faveolata* exhibits strong overall connectivity (Fst = 0.038). However, this overall connectivity was complicated by regional patterns of genetic structure, including an east-west genetic barrier at Mona Passage and differentiation across only 470 Km along the Mesoamerican Barrier Reef System (*Rippe et al., 2017*).

In addition to connectivity of the coral host, our results agree with previous work demonstrating greater genetic differentiation of symbiont populations. For example, population differentiation of algal symbionts of the genus *Cladocopium* (C3, C40) associated with *A. hyacinthus* and *A. digifiera* across Micronesia occurs across smaller scales compared to the coral animal (*Davies et al., 2015*; *Davies et al., 2020*). While the coral hosts exhibited gene flow across 4,000 Km of the Pacific (*Davies et al., 2015*), the algal symbiont populations were often diverged between reefs within the same island (*Davies et al., 2020*). *Pettay & LaJeunesse (2013)* found a similar pattern in *Durusdinium glynni* population structure sampled from *Pocillopora* (type 1) across the Eastern Tropical Pacific. In contrast to the host, which exhibited genetic connectivity over 3,400 Km (*Pinzon & LaJeunesse, 2011*), a subtropical population of *D. glynni* in the Gulf of California was differentiated from all other populations, the closest being approximately 700 Km away (*Pettay & LaJeunesse, 2013*). This pattern of limited symbiont population connectivity compared to its coral host has also been shown in *A. palmata* hosting *Symbiodinium fitti* in the Caribbean (*Baums, Devlin-Durante & LaJeunesse, 2014*), symbionts of the genus *Cladocopium* hosted by the octocoral *Sinularia flexibilis* on the Great Barrier Reef (*Howells, Van Oppen & Willis, 2009*), symbiont haplotypes B1/B184 associated with corals of the genus *Montastraea* (*Thornhill et al., 2009*), and in octocorals *Gorgonia ventalina* (*Kirk et al., 2009*) and *Pseudopterogorgia elisabethae* (*Santos et al., 2003*) associated with symbionts of the genus *Breviolum*. Taken together, these studies on both scleractinian and gorgonian tropical corals reveal that limited connectivity of Symbiodiniaceae in comparison to the coral host is a commonly observed pattern. This suggests distinct micro-evolutionary processes (e.g., lower effective dispersal in the symbiont or stronger drift of symbiont genotypes within coral colonies) affect each symbiotic partner (*Baums, Devlin-Durante & LaJeunesse, 2014*). Interestingly, our data confirm that pattern in a temperate coral species that is only facultatively associated with algal symbionts, thus the factors limiting dispersal of Symbiodiniaceae appear unrelated to the degree of host fidelity in the symbiosis.

## Physical patterns of water masses could contribute to the connectivity of *A. poculata* hosts

Population structure of organisms whose range spans the east coast of the United States is influenced by the physical properties of the ocean, where the warm waters of the Gulf Stream current meet the cooler, less-saline Labrador Current to create one of the steepest gradients in latitudinal temperature change in the world (*Bower, Rossby & Lillibridge, 1985*; *Conover et al., 2006*; *Wares, 2002*). Along this latitudinal gradient, Cape Hatteras and Cape Cod are sites of particularly stark environmental transitions (*Wares, 2002*). Cape Hatteras separates warmer waters in the south (the Carolinian Province) from more temperate and seasonally fluctuating waters to the north (Virginian Province; *Mach et al., 2011*), while Cape Cod separates the Virginian Province from the consistently cooler waters of the Acadian Province (*Briggs, 1974*; *Engle & Summers, 1999*; *Mach et al., 2011*). Cape Cod also represents the upper range limit of *A. poculata* (*Peters et al., 1988*; *Thornhill et al., 2008*), where it is likely restricted by the colder waters to the north (*Dimond et al., 2012*). Both VA and RI populations considered here lie within the Virginian Province and therefore between these stark environmental breaks (Fig. 1). This likely facilitates dispersal of *A. poculata* larvae along the coast between sites, where they might otherwise be prevented from dispersing by the Gulf Stream to the south or around Cape Cod to the north.

These biogeographical provinces of the northwest Atlantic influence population structure not just of *A. poculata*, but of a diverse collection of organisms. For example, using RAD-seq, *Boehm et al. (2015)* found two populations of the lined seahorse (*Hippocampus erectus*) within the Virginian Province to be connected and from an ancestral gene pool that diverged from populations south of Cape Hatteras in the Carolinian and Caribbean provinces. In contrast to panmixia within the Virginian Province, as we also observed in the *A. poculata* host, *Zhang, Geller & Vrijenhoek (2014)* sequenced two nuclear genes (ANT and H3) and found a break in structure in populations of the amethyst gem clam (*Gemma gemma*; no planktonic larval phase) around New Jersey, separating southern populations of Maryland, Virginia, and North Carolina from northern populations of Maine, Massachusetts, and Connecticut. Such divisions in population structure between what are called the Upper and Lower Virginian provinces (at ~39°N latitude) correspond with a gradient in average SST and with population structure of several other species, including amphipods (*Ampithoe longimana*), killifish (*Fundulus heteroclitus*), polychaetes (*Marenzellaria viridis*), and copepods (*Eurytemora affinis*; reviewed in *Wares (2002)*). This division between Upper and Lower Virginian provinces is more consistent with what we observe in *B. psygmophilum* population structure. In contrast to a body of literature demonstrating marine species with phylogeographic structure in the northwest Atlantic, including those highlighted above, *Strasser & Barber (2009)* found no evidence of genetic structure in the softshell clam (*Mya arenaria*) across the stark environmental differences between Maryland in the south and Nova Scotia in the north by sequencing the mitochondrial cytochrome oxidase I gene. The authors suggest that high levels of dispersal and gene flow, facilitated by a 3 week

planktonic larval phase, likely contribute to the observed pattern (*Strasser & Barber, 2009*).
For these studies mentioned above, it is possible that differences in the genetic markers
used to look for population structure also influenced the ability to detect differences
(*D'Aloia et al., 2020*). In any case, this body of work highlights that oceanographic
currents, planktonic larval durations and behavior, life history dynamics, and the
environment are all important to consider when evaluating population structure in the sea.

## Temperature as a potential driver of adaptive population differentiation

Although there are no severe environmental breaks between VA and RI, as both sites
lie within the Virginian Province, we have previously shown that there are indeed
differences in the temperature environments at our collection sites, including warmer
summers in VA and colder winters in RI (*Aichelman, Zimmerman & Barshis, 2019*).
These temperature differences are not restricted to the time frame of this study, but are
consistent over the last ~40 years of SST data (Fig. 1). These environmental differences
could drive the subtle adaptive divergence we found in the *A. poculata* host as well as
the *B. psygmophilum* symbiont. There is a large body of evidence demonstrating that
differences in temperature contribute to adaptive population differentiation in marine
organisms across a variety of spatial scales (reviewed in *Sanford & Kelly (2011)*).
For example, *Matz et al. (2018)* found population differentiation of *A. millepora* on the
Great Barrier Reef across over 1,200 Km was associated with temperature. In a model
simulation, this work showed that even high migration rates did not interfere with patterns
of local thermal adaptation, and that the metapopulation could be able to adapt to
predicted warming over the next 100 to 250 years (*Matz et al., 2018*). Additionally,
*Haguenauer et al. (2013)* found that populations of the Mediterranean octocoral *Corallium
rubrum* from distinct thermal regimes across depth (5, 20, 40 m) differentially induced
heat shock protein 70 (HSP70) expression as a function of thermal history. Namely,
shallow-water corals that historically experienced warmer temperatures more strongly
induced expression of HSP70 upon heat stress, and microsatellite loci showed that corals
from the three depths were genetically differentiated across this steep environmental
gradient (*Haguenauer et al., 2013*).

Gradients in environmental parameters such as temperature also play an important role
in local adaptation (*Pettay & LaJeunesse, 2013*) and genetic diversification (*LaJeunesse
et al., 2014*) of Symbiodiniaceae. *Pettay & LaJeunesse (2013)* showed differentiation of
the subtropical population of *D. glynni* symbionts corresponds to environmental
differentiation, including seasonal variation in temperature and light. It was hypothesized
that connectivity in the *Pocillopora* coral host plus differentiation in the *D. glynni* symbiont
resulted from larvae and associated symbionts arriving from more southern reefs to
the Gulf of California being rapidly replaced by symbionts better adapted to the local
temperate environment (*Pettay & LaJeunesse, 2013*). In contrast to *A. poculata*, *Pocillopora*
maternally inherit their symbionts (vertical transmission). However it is possible that,
similar to *Pettay & LaJeunesse (2013)*, differences in seasonal temperature (here between
VA and RI; *Aichelman, Zimmerman & Barshis, 2019*) are driving local adaptation in the
*B. psygmophilum* symbionts that *A. poculata* larvae acquire upon settling in their

respective environments. Additionally, in the facultatively symbiotic coral *Oculina patagonica*, *Leydet & Hellberg (2016)* show that distinct symbiont communities across the corals' Mediterranean range correlated with sea surface temperature rather than host genetic background, supporting our hypothesis that local environment also plays an important role in symbiont community structure of facultative corals.

## Outlier SNPs related to coral stress response and energetics

It is important to distinguish between neutral and adaptive loci when considering population divergence. Namely, loci can still be under selection even in the face of extensive gene flow, and therefore populations can appear homogeneous at neutral loci but still exhibit local adaptation (*Conover et al., 2006*). Adaptive (as opposed to neutral) genetic variation affects an organism's fitness (*Conover et al., 2006*). Here, we find evidence of *A. poculata* host connectivity at neutral loci despite evidence for adaptive divergence at several high outlier sites.

Several of the putatively adaptive SNPs found here occur on genes that have been previously associated with the coral stress response. For example, two putatively adaptive SNPs were each found on genes annotated as 40S ribosomal protein S3 and 60S ribosomal protein L26. Both 40S and 60S ribosomal proteins were shown to be downregulated in *Acropora palmata* after 2 days of thermal stress (~32 °C; *DeSalvo et al., 2010*). Additionally, another four putatively adaptive SNPs were found on the gene annotated as myosin heavy chain. Myosin heavy chain is associated with the cytoskeleton and was previously reported to be upregulated after 24 h of thermal stress (28 °C) in the octocoral *Scleronephthya gracillimum* (*Woo et al., 2010*). Two more putatively adaptive SNPs were located on the gene annotated as ubiquitin-like protein FUBI. Ubiquitin binds to damaged proteins and marks them for degradation and reuse, and ubiquinated proteins are therefore thought to be a key marker for the physiological stress response (*Weis, 2010*). *Barshis et al. (2010)* found that ubiquitin-conjugated proteins were constitutively higher in a more thermally tolerant back reef population of *Porites lobata* in American Samoa. Lastly, although not related to coral response to thermal stress, two putatively adaptive SNPs were found on a gene annotated as apolipoprotein B-100, which *Bertucci et al. (2015)* suggest plays a role in the metabolite exchange important for the coral-algal symbiosis. It is therefore possible that adaptive differentiation between VA and RI *A. poculata* populations is occurring at loci related to the coral stress response and/or symbiosis.

We found evidence of four putatively adaptive loci in *B. psygmophilum*, three of which were annotated, including photosystem II (PSII) CP43 reaction center protein (*psbC* gene; binds chlorophyl and helps catalyze light-induced photochemical processes of PSII), photosystem I (PSI) P700 chlorophyl *a* apoprotein A2 (*psaB* gene; reaction center chlorophyl *a* molecule in association with PSI), and PSII protein D1 (*psbA* gene; core of the PSII reaction center). Species of Symbiodiniaceae have been previously shown to be differentiated by mutations on genes related to photosystem machinery. *Wham, Ning & LaJeunesse (2017)* found that *Durusdinium glynnii* was distinguished from other symbionts within the genus *Durusdinium* as well as from symbionts of other genera by a

non-synonymous mutation that effected a change on the D1 protein of photosystem II (*psbA* gene), a highly conserved region in dinoflagellates (*Iida et al., 2008*). This could indicate that differences in temperature and/or light environment between VA and RI are driving adaptive differentiation of *B. psygmophilum* populations at loci related to photosynthetic machinery. Future work should aim to further characterize the differentiation of *B. psygmophilum* across the latitudinal range of *A. poculata*, for example by sequencing other pertinent regions of the genome (i.e., chloroplast-encoded 23S rDNA or internal transcribed spacers) or considering cell size differences (*LaJeunesse, Parkinson & Reimer, 2012*; *LaJeunesse et al., 2014*).

Additionally, as there appears to be some differentiation between white and brown corals based on the PCA analysis (Fig. 2B) and within population pairwise Fst calculations, it is tempting to speculate that there are factors driving genetic divergence between symbiotic states. By considering each population separately and looking for consistent outlier SNP identities, we do not find compelling evidence to support this hypothesis as only 11 out of 49 (for VA) and 47 (for RI) outlier SNPs were shared between the two groups. Thus, we find it less parsimonious that each population has uniquely evolved genetic differentiation between symbiotic states, as we imagine a high degree of conservation/shared evolutionary history in this trait. However, there is one interesting high outlier SNP in both VA and RI, annotated as Sequestosome-1, that could potentially be related to differentiation between symbiotic state. Sequestosome-1 (UniProtKB entry 008623; *The UniProt Consortium, 2018*) is involved in autophagy, and may regulate the activation of NF-κB1 (a conserved immune regulatory protein) by TNF-alpha. Previous work in a sea anemone model by *Mansfield et al. (2017)* suggested that NF-κB levels were related to symbiotic state, and Symbiodiniaceae suppressed NF-κB to establish symbiosis with *Exaiptasia pallida*. While it is possible that our results could signify a host-specific genotypic difference that is associated with being symbiotic or aposymbiotic, it is far from definitive. The environmental and/or genetic factors determining which *A. poculata* individuals are symbiotic and which are aposymbiotic therefore remains elusive, and certainly warrants further exploration.

For both the coral host and symbiont outlier loci, it is important to acknowledge the possibility that some of the SNPs we discuss above are false positives, and not true signatures of local adaptation. Additionally, it will be important to conduct deeper sequencing and analyze additional loci across multiple populations to have more confidence in the outlier loci driving differentiation across the range of *A. poculata*.

## CONCLUSIONS

Here, we found contrasting levels of genetic connectivity in the different partners of the *Astrangia poculata* holobiont, with neutral gene flow and adaptive divergence in the coral host versus neutral and adaptive divergence in the algal symbiont. This highlights how the interacting forces of oceanography, environmentally driven selection, local adaptation, and reproductive biology can manifest in differential connectivity of a marine holobiont system. Previously, we found physiological differentiation in coral host

metrics (i.e., respiration rate and thermal optima), but not in symbiont physiology (i.e., photosynthesis rate and photochemical efficiency; *Aichelman, Zimmerman & Barshis, 2019*). Interestingly, here we find the opposite, namely a stronger signal of divergence in the algal symbiont. It is possible that our previous hypothesis regarding symbiont acclimation to aquarium light conditions accounts for this discrepancy, but future work should aim to disentangle the possibility for local adaptation acting at different levels of the *A. poculata* holobiont. Although *A. poculata* inhabits hard bottom communities throughout the mid-Atlantic, much remains to be learned about the physiological and molecular mechanisms that facilitate its persistence in such extreme temperature ranges, particularly outside the more northern range focus of this study. Future work should explore extended sampling, and include deeper sequencing to improve the search for potentially selected alleles, across the strong potential biogeographic break of the Gulf Stream at Cape Hatteras as well as increasing the spatial resolution to characterize the scale of connectivity in the weakly dispersing symbionts. Such work will help elucidate the potential role of temperature in driving local adaptation of *A. poculata* and *B. psygmophilum*, and could provide insights into how the population dynamics of this holobiont could change as temperatures continue to warm.

## ACKNOWLEDGEMENTS

We extend appreciation to K. Sharp, R. Rotjan, S. Grace and the annual *Astrangia* Workshop hosted by Roger Williams University for fostering creative conversations and collaborations leading to this work. We thank Sandrine Boissel and Dr. Courtney Klepac for assistance with field collections and lab work and Dr. Hanny Rivera for assistance with STRUCTURE and other beneficial discussions regarding data analysis. Also, thanks to Kristina Bounds and Matt Bengtson for assistance with aquaria maintenance. This research was supported by the Research Computing clusters at Old Dominion University, which supported all of the computing resources that were used to analyze the data.

### Funding

This work was supported by a Virginia Sea Grant Graduate Research Fellowship (to Hannah E. Aichelman and Daniel J. Barshis, No. R/71856J) and a PADI Foundation Grant (to Hannah E. Aichelman). The funders had no role in study design, data collection and analysis, decision to publish, or preparation of the manuscript.

### Grant Disclosures

The following grant information was disclosed by the authors:
Virginia Sea Grant Graduate Research Fellowship: R/71856J.
PADI Foundation Grant.

### Competing Interests

The authors declare that they have no competing interests.

## Author Contributions

- Hannah E. Aichelman conceived and designed the experiments, performed the experiments, analyzed the data, prepared figures and/or tables, authored or reviewed drafts of the paper, and approved the final draft.
- Daniel J. Barshis conceived and designed the experiments, analyzed the data, authored or reviewed drafts of the paper, and approved the final draft.

## Field Study Permissions

The following information was supplied relating to field study approvals (i.e., approving body and any reference numbers):

Virginia coral collections were conducted under the Virginia Marine Resources Commission permit #17-017. No permit was required for collection of Rhode Island corals.

## DNA Deposition

The following information was supplied regarding the deposition of DNA sequences:

All sequencing data are available at NCBI BioProject: PRJNA614998.

## Data Availability

All analyses, scripts, and Supplemental Files are available at GitHub: https://github.com/hannahaichelman/Astrangia_PopGen.

## Supplemental Information

Supplemental information for this article can be found online at http://dx.doi.org/10.7717/peerj.10201#supplemental-information.

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
