# Peer review of "Adaptive divergence, neutral panmixia, and algal symbiont population structure in the temperate coral Astrangia poculata along the Mid-Atlantic United States"

_PeerJ, doi:10.7717/peerj.10201_

## Round 0.1 · original submission · Major Revisions

Two expert reviewers have evaluated your manuscript and their in-depth reviews can be seen below. Both reviewers are positive with respect to the relevance of this study, however, they also have made some important suggestions and comments. In particular, the materials and methods section should be more detailed and the limitations of the study should be addressed. These include the low SNP number, that corals were placed in one warm and one cold tank thus limiting replicates, and to answer whether the samples were tested for clones. Also, this dataset provides an excellent opportunity to discuss the brown and white morphs.

Reviewer 1 ·

Basic reporting

In this manuscript (#47962), entitled “Adaptive divergence, neutral panmixia, and algal symbiont population structure in the temperate coral Astrangia poculata along the Mid-Atlantic United States”, Aichelman and Barshis aim to address a central question in marine molecular ecology: what are the forces shaping the pattern of genetic structure within species?
Focusing of a temperate and facultative symbiotic coral, the northern star coral, Astrangia poculata (= A. danae), they “use high-throughput mRNA sequencing to consider patterns of neutral and adaptive differentiation” in the host and the symbiont, Breviolum psygmophilum (l.101-103). More particularly, they “characterized population structure of the A. poculata host and its symbiont (B. psygmophilum) both by population (VA vs. RI) and by symbiotic state (brown vs. white) using SNPs derived from mRNA-Seq data” (l.121-123). Interestingly, this work complements a recently published paper (Aichelman et al. 2019) suggesting the occurrence of adaption to local thermal conditions in this species.
The Authors focused on symbiotic and asymbiotic individuals from two populations: “10 brown and 10 white Astrangia poculata colonies were collected each from Virginia (VA) and Rhode Island (RI), USA” (l. 130-131). These individuals were submitted to a common garden experiment (l.165-204) “designed to consider differential gene expression in A. poculata across temperatures and populations” (l.152-153). Here, the Authors present “the sequencing data from this experiment and the resulting single nucleotide polymorphisms (SNPs) to consider population structure” (l.153-155).
The Authors sequenced 96 libraries (84 for gene expression analyses + 12 for population genetics analyses) from 10 symbiotic and 10 asymbiotic individuals from two localities. They conducted the de novo transcriptome assembly and SNP calling and filtering. They analyzed the population genetic structure contrasting the pattern between “neutral” and “outlier” loci in the host and the symbiont using an AMOVA, F-statistics and PCA (l.272-315). In addition to gene ontology analyses was conducted on outlier SNPs to “determine if genes containing outlier SNPs were enriched for any particular GO categories” (l.318).
Based on these analyses, the Authors discuss: 1) the occurrence of contrasting patterns of genetic structure among the host and the symbiont; 2) the potential impact of physical barrier on connectivity among the two populations; 3) the influence of local temperature regime on the pattern of genetic differentiation; 4) identify putative genes that may be involved in adaptation to local thermal conditions.



Basic reporting:

The paper is well written (except some typos, eg. populationS on l. 332, 333).

Ideas are clear, reasoning is well exposed.

The Introduction section gives an interesting perspective starting with general considerations on marine connectivity, issues linked to the study of holobionts and the potential role of temperature on the pattern of intra-specific diversity. The model and the objective of the study are well presented and described. One potential flaw may be the lack of reference to works already published in temperate corals (except Costantini et al. 2007). The Authors may consider the following papers focused: i) on the neutral genetic structure in temperate corals: Ledoux et al. 2010 Mol. Ecol., Ledoux et al. 2018 JBiogeo, Aurelle et al. 2011 Genetica, Arizmendi-Meija et al. 2015 Plos One, Masmoudi et al. 2016 EcolEvol; ii) on the potential for local adaptation to thermal conditions in temperate corals: Ledoux et al. 2015 Ecol Evol, Arizmendi-Meija et al. 2015 Coral Reefs, Haguenauer et al. 2013 JEMBE, Crisci et al. 2017 Scientific Rep. The Introduction and the Discussion sections may be strengthened by adding some of these references.

In addition, one of the main originalities of the paper is to be focused on a facultative symbiotic coral species. Whether these models have been previously studied in the context of population genetics or local adaptation remain unclear based on the Introduction. This may be improved.

Details on the previous work conducted by the Authors (Aichelman et al. 2019) may be mentioned in the Introduction instead of being presented in the Material and Methods.

The structure of the paper is conformed to PeerJ Standards. Figures and tables are relevant. I would suggest to complete the Figure 1 with the localities discussed from l. 428-468 (e.g. Cap cod, Virginia province).

I probably missed them but the raw data do not seem to be available (no depository number).

Experimental design

Experimental design:
The issue addressed by the Authors perfectly fits the scope of PeerJ.

The research questions are relevant. However, it would be nice to explain why do the Authors focused on the potential impact of temperature in this particular species (e.g. potential impact of climate change?).

Investigation is rigorous in line with high ethical and technical standard.

The methods are sufficiently detailed in most of the manuscript. However, I have some minor concerns regarding the design of the common garden experiment as well as the filtering of molecular markers.
In my understanding, the idea of common garden would be used to name the whole experimental design (e.g. Villemereuil et al. 2015 Heredity; Kawecki 2004 Ecology Letters) and not only the holding tank as did by the Authors. This may be confusing and should be rephrased.
The Authors did not replicate the experimental conditions (i.e. only one warm and one cold tank). Logistic may be a limiting factor for replication, however this lack of “experimental tank” replication and potential logistical limitations should be mentioned.

Did the Authors genetically confirm the individuals used in the experiment were not clone?

Regarding the SNP markers, the Authors conducted the SNP filtering at two levels: considering two and four populations, respectively. It is not clear from the manuscript why the Authors choose this approach resulting in two different sets of SNPs, instead of considering the same set of SNPs for two and four populations and to what extend the SNPs that are not shared among the two levels influence the results of the analyses. This should be considered by the Authors.

The Authors stated “It should be noted that the dataset presented here had relatively low read counts, resulting from an error in the library preparation that led to sequencing of rRNA in addition to mRNA. To combat this, we have used detailed screening methods at every step of the data analysis pipeline to address these issues” (l.277-280). While they mentioned “detailed screening methods”, I was not able to identify these particular screening methods in the present version of the MS. Moreover, the potential implications of the errors in library preparation should be explained.

The number of filtered SNPs is relatively low (1,506 and 1,562 considering 2 and 4 host populations, respectively and 68 for the symbiont). From these pools of SNPs and following the Baypass results, the analyses were conducted on 226 and 9 neutral and selected SNPs and 213 and 12 neutral and selected SNPs for the four and two populations analyses, respectively.
The difference in the number of high-quality SNPs and the SNPs used for analyses should be explained both for the host and the symbiont.

It may have been interesting to consider the whole pool of SNPs first (neutral + selected) and then to conduct the analyses on neutral vs. potentially selected markers, separately.

The Authors mentioned “high-quality SNPs”. What do the “high-quality” refer to?

Moreover, considering the high dispersal capacity of Astrangia poculata, the expected level of genetic structure should be very low. Accordingly, the lack of significant genetic structure reported from neutral SNPs may potentially be linked to the low number of SNPs analyzed. Did the Authors consider this hypothesis?

Validity of the findings

Validity of the finding:
Overall, this is an interesting study focused on an original model (temperate facultative symbiotic coral). Conclusions are globally well supported by the analyses and fully linked to the research question.

I would suggest the Authors to rephrase parts of the manuscript addressing the contrasted patterns of genetic differentiation among the host and the symbiont to better fit with the analyses and results. Indeed, the pattern of “selected” genetic structure was not characterized in the Symbiont.

The Authors may also consider to precise the type of genetic markers used when discussing the results of previous population genetics studies conducted in the same region (paragraph 4.2).

It would be interesting to develop the idea that selection may drive the pattern of differentiation at the neutral markers in the Symbiont (l.476).

The Authors stated “Interestingly, the stronger signal of divergence in the algal symbiont is in contrast with our previous results demonstrating the strongest physiological population divergence in coral host respiration” on l. 541-543. In the present form, this sentence may appear a bit vague. The Authors should consider to develop a bit more this issue and its implication in term of local adaptation both at the symbiont and host levels.

Additional comments

General comments:
As mentioned, the study presented by Aichelman and Barshis is interesting and well written. In my opinion, it fits the standard of publication in PeerJ. However, I encourage the Authors to account for the comments previously made in order to clarify some parts of the paper.

Reviewer 2 ·

Basic reporting

This article is clearly presented, and well organised. It is clearly written.

Experimental design

This article presents original results on the genetic structure of a temperate scleractinian coral and its associated dinoflagellate symbiont in two locations of western Atlantic. The research question is meaningful. The data come from mRNA sequencing which was initially performed for an experimental study of thermotolerance. The results are based on an analysis of potentially selected loci, on differentiation tests and Fst estimates, and on multivariate analyses.

The results are interesting, but their interpretation is limited by the quite low final number of SNPs retained in the study, especially for the symbiont. This is a problem for the search of potentially selected alleles, and this point should be in the discussion: is it really meaningful to test for outlier loci with only 68 SNPs? The study is also based on the comparison of only two populations which also limits the conclusion about local adaptation, and replicates would be useful. I understand this is not possible for the present study, but this point should be discussed as well.

Validity of the findings

From a methodological point of view, the authors tested for outlier loci in their dataset. I wonder if this analysis can be biased for dinoflagellates considering their life cycle and ploidy ? For the outlier analyses there was no filtering to keep one SNP per locus : why ? We can see on line 357 that nine coral contigs contained more than one outlier : I wonder if this could have modified the analysis. More generally, in the SNP filtering the authors did not retain SNPs which were not in Hardy-Weinberg equilibrium : but for dinoflagellate, do we expect panmixia ? For the outlier SNPs, both for coral and symbionts, do they correspond to non-synonymous mutations ? And I think that analyses of outlier loci should include several different methods to check the robustness of the results.
Regarding the coral samples, it would be necessary to indicate if putative clones were detected in the dataset (as this can bias the study of genetic structure). Apart from the genetic structure, the authors could give some estimates of genetic diversity such as expected heterozygosity. For this coral dataset, an analysis of genetic structure (with the software STRUCTURE or another similar method) would be interesting to estimate levels of admixture in individuals. There is also an interesting result of differentiation among white and brown coral morphs in Rhode Island (see Fst and PCA), but this is not discussed. More generally the white / brown differentiation is not discussed here, whereas this dataset is a very good opportunity to study this polymorphism.

Additional comments

Abstract :
"In the symbiont, the two putatively adaptive loci…" (there are only two candidate loci, right?)

Introduction :
line 69 : factors instead of forces
line 78 : some organisms or all organisms ?
line 94 : as drivers
lines 95-97 : maybe give some species names
lines 99-100 : you could check the litterature on the thermotolerance of Mediterranean octocorals for example
line 103 : at the beginning of the sentence, write the full genus name
line 109 : give some numbers
line 119 : five weeks in aquarium right ?
line 121 is similar to line 101


Materials and methods :
lines 130-134 : were brown and white colonies sampled exactly at the same site and in the same ecological conditions ?
line 132 : samples sizes are already given on line 130
line 140 : is there a possibility to collect colonies issued from asexual propagation ?
lines 174-181 : what about light and food ? Did you use the same conditions for white and brown colonies ?
line 194: the experiment was a short stress, but the sampling lasted 1 h 40 min after the experiment : please explain this (e.g. what were the first and last samples?).
lines 254 – 255 : do the numbers refer to the number of species in the different databases ?
line 263 : this filtering seems logical, but then you can’t test here if there was some signals of dinoflagellate sequences in white morphs ?
line 296 : what is the meaning of "h" in your Hardy-Weinberg filtering ?
lines 300-302 : give a brief explanation about how this method works. Why didn’t you filter for on SNP par locus here ?
And it would be necessary to test other methods to detect loci under selection
line 303 : in table 3 you mention low and high outliers : please explain here.

Results :
line 330 : by "sites" do you mean bp ? (because vcf files usually refer to variant sites)
lines 332-333 : recall here what are the two and four populations analyzes (and use plural for populations)
lines 344-345 :it would be interesting to analyse other axes in PCA (this can be in supplementary material)
line 353 : " a total of 19 sites were used" : because you test later if there are some outliers, you can’t say here that they are neutral
line 353-354 : indicate that the two outliers were not on the same contig (deduced from supplementary material but this should be mentioned here). And you could nevertheless do an Fst analysis on this SNPs.

Discussion :
line 388 (see also line 407) : you did not describe patterns of connectivity here but patterns of genetic differentiation. For the dinoflagellate, the genetic differentiation could also be driven by drift of different genotypes inside colonies which could lead to genetic differentiation among populations. This could also be driven by selection of different genotypes even if this was not detected by your outlier analysis because of a low number of SNPs.
line 419 : Sinularia is an octocoral
line 421 : I would suggest octocoral rather than soft coral
line 449 : "connected and isolated" ? I don’t understand
line 465 : what about the larval duration for the other species mentioned above ?
line 475-476 : please explain how environmental differences could drive divergence in neutral loci.
line 482 : "the metapopulation could be able" rather than "was able"
line 493 : "similar differences" : similar to what ?
lines 501-502 : for this kind of analysis it is also important to consider and discuss the possibility of false positives.
line 518 : response to which stress then ? You just mentioned copper. And you also mentioned a gene putatively involved in variation in symbiont density (so maybe not only coral response).
lines 529-533 : it would be important to analyze more loci to get more confidence in outlier analysis

Conclusion
line 542 : why do you not discuss this ?

References : check the bibliography format. For example some references use abbreviated journal name, and some others use full name

---

## Round 0.2 · accepted · Accept

I am satisfied with the improvements that have been made to the manuscript.

Reviewer 1 ·

Basic reporting

no comment

Experimental design

no comment

Validity of the findings

no comment

Additional comments

This is the second time I review (Reviewer #1 in the previous round) the manuscript entitled “Adaptive divergence, neutral panmixia, and algal symbiont population structure in the temperate coral Astrangia poculata along the Mid-Atlantic United States” by Aichelman and Barshis.

First of all, I would like to acknowledge the impressive work made by the Authors during the revision.They addressed and clarified most of the topics raised from the first version of the MS and I believe the manuscript has significantly improved.

I thus recommend the manuscript for acceptance.